# Experimental Analysis of the Magnetic Leakage Detection of a Corroded Steel Strand Due to Vibration

**DOI:** 10.3390/s23167130

**Published:** 2023-08-11

**Authors:** Hong Zhang, Yaxi Ding, Ye Yuan, Runchuan Xia, Jianting Zhou

**Affiliations:** 1State Key Laboratory of Mountain Bridge and Tunnel Engineering, Chongqing Jiaotong University, Chongqing 400074, China; 622210086048@mails.cqjtu.edu.cn (Y.D.); yuanye202307@163.com (Y.Y.); rcxia@mails.cqjtu.edu.cn (R.X.); jtzhou@cqjtu.edu.cn (J.Z.); 2School of Civil Engineering, Chongqing Jiaotong University, Chongqing 400074, China

**Keywords:** stranded steel, cable structure, vibration, corrosion, self-magnetic flux leakage

## Abstract

The self-magnetic flux leakage (SMFL) detection technique has great potential in the corrosion detection of bridge stay cables due to its advantages of small testing equipment, high accuracy, and fast testing rate. However, the vibration effect in the cable’s SMFL detection is unclear. To address this, the influence of vibration on the magnetic field distribution of cable structure is analyzed theoretically. According to the theoretical model, the effect of vibration on SMFL detection primarily manifests as displacement changes (displacement-added magnetic field) and defect shape changes (deformation-added magnetic field). SMFL detection experiments are conducted on steel strands. The results demonstrate that the displacement-added magnetic field exhibits statistical characteristics in the form of a normal distribution, fluctuating around the zero value. The impact of the deformation-added magnetic field on SMFL is linearly correlated with the corrosion ratio *c.* Moreover, a corrosion characterization index *A* was proposed and has an excellent linear fit with the corrosion ratio *c*. The index *A* effectively improves the accuracy of corrosion detection and provides early warning for the maintenance of cable structures.

## 1. Introduction

### 1.1. Motivation

In recent years, cable-stayed bridges were rapidly promoted and applied worldwide due to their beautiful appearance, high-crossing capacity, and relatively low construction cost [1]. As the primary force-transmitting component of cable-stayed bridges, the performance of the cable directly affects the reliability of the bridge’s structural system [2,3,4,5]. Due to the prolonged exposure of cables to external conditions, they are susceptible to various environmental factors, such as ultraviolet radiation, high temperature and humidity, acid rain, as well as the combined effect of cyclic loads. As a result, different degrees of damage, such as protective coating erosion, water penetration, and corrosion or breakage of internal steel wires, can occur in the cable system [6,7,8,9]. Existing research and engineering practices showed that the performance degradation of a cable often begins with sheath breakage and corrosion of high-strength steel wires [10,11]. Therefore, the research on diagnostic techniques for assessing the service condition of cables was a focal point of attention in both the engineering and academic communities [12].

Non-destructive testing methods became the fastest-growing inspection method due to their ability to assess structural conditions without causing direct damage [13]. The main non-destructive testing methods for detecting defects in cable structures include ultrasonic-guided wave testing [14], acoustic emission testing [15], magnetostrictive-guided wave testing [16], and magnetic leakage testing [17]. In addition, the cables’ internal steel wires/strands are ferromagnetic materials, and the magnetic leakage phenomenon occurs in the damaged areas. The magnetic leakage signal in these areas is not hindered by the PE sheath of the cable and exhibits a strong correlation with cable corrosion and wire breakage.

Therefore, among the various non-destructive testing methods for cable structures, the self-magnetic flux leakage (SMFL) testing technique is increasingly recognized and valued by experts and scholars due to its advantages of small testing equipment, high accuracy, and fast testing rate [18,19]. However, this method is prone to interference from spatial magnetic fields, the PE sheath, and external environmental factors [20]. Therefore, it is of great scientific significance and potential engineering application prospects to research the spontaneous magnetic leakage testing method for detecting wire corrosion in cable structures under the coupling effect of multiple factors in natural bridge conditions [21,22,23].

### 1.2. Literature Review

In the past few decades, SMFL testing for wire corrosion in cables received significant attention from experts and became a research hotspot. The research focus was primarily on the characterization mechanism of magnetic leakage, diagnostic methods for wire corrosion, magnetic signal acquisition, analysis methods, as well as studies on SMFL imaging and inversion methods [24]. For example, Shi [25] reviewed the mechanisms of the magnetic memory effect, influencing factors of magnetic signals, and evaluation methods for magnetic memory damage over the past decade. Qiu [26] derived the theoretical framework for SMFL testing of parallel steel wires/strands, based on the magnetic dipole model. Xia [27] simplified the defective region as a sector shape for the case of continuous corrosion of multiple wires in the outer layer of parallel wire ropes. They derived a three-dimensional spatial model for spontaneous magnetic leakage by combining magnetic charge density theory and the magnetic dipole model [28].

Experts and scholars have conducted numerous basic experiments while conducting spontaneous magnetic leakage theory characterization research. For example, Fernandes found that the corrosion region evaluated with the SMFL typically matched the actual corrosion observed by visual inspection. They used SMFL to detect the remaining cross-sectional area of a corroded prestressed steel strand [29]. Qu [30] proposed a method of quantitatively evaluating the corrosion width of a steel bar using the intersection point of SMFL curves. They offered a novel way of predicting corrosion width with better accuracy. Singh [31] developed a flexible giant magnetoresistance (GMR) sensor array for the magnetic leakage detection of steel wire ropes, and conducted preliminary binary image imaging exploration of wire break defects. Ren [32] studied the effect of initial magnetization on the magnetic memory signal for the quantitative testing of SMFL detection. Xia [33,34,35] developed a logistic model to analyze the eigenvalue of the SMFL signal and thus quantitatively assessed the corrosion of steel strands. Shi [36] determined the effects of the load, defect size and lifting height on the magnetic signals. A theoretical model was applied to solve stress concentration problems in demonstrating the feasibility of the model in early diagnosis. Le [37] proposed a method of simulating the magnetic field distribution in magnetic flux leakage detection using a deep neural network. In terms of quantitative characterization of corrosion morphology, Xu [38] developed an indirect method for describing the corrosion morphology by injecting 60 mL of red ink into the cracked sheath of the old cable removed from a bridge and observing the distribution of the red ink within the cable body after two days. Additionally, Li [39] utilized a high-resolution three-dimensional contour measurement device to obtain geometric point cloud images of the steel wire surface, and through digital image processing techniques they derived relevant statistical features of the corroded pits on the steel wire surface. This method provides a feasible and reliable approach for accurately quantifying the corrosion morphology.

Inspired by the literature review, the diagnosis of corrosion and broken wire status in tensioned suspension cables based on the spontaneous leakage magnetic detection technique is feasible, and significant progress is made in existing research. While some challenges and issues still need to be addressed, the above research on detecting the corrosion of cable structures using SMFL technology mainly focused on the testing technique. However, the effects of the environment on natural structures were ignored. The high accuracy of SMFL detection technology, for instance, is easily affected by the external environment [40,41]. The cable undergoes continuous vibrations due to alternating loads, including the vibrations caused by bridge structures, vehicle loads, wind-induced vibrations [42,43], and rain-wind-induced vibrations [44]. The deformation of the cable structure differs from its static state, which inevitably influences corrosion measurement [45], as well as the stability and accuracy of the data. Precisely measuring the corrosion of bridge cables has become an urgent task for making maintenance decisions. Analyzing the impact of vibration on the measurement is also of critical scientific research significance. Therefore, the paper aims to study the influence of the vibration on the SMFL detection of cable corrosion, analyze its influencing mechanism, and find effective methods to reduce the impact of the vibration.

### 1.3. Contribution

In this study, the vibration characteristics of the corrosion distribution of cable structures using SMFL technology were theoretically interpreted. Both vibration simulations and SMFL tests were performed on corroded steel strands through experimental analysis. The characteristics of the magnetic field distribution under vibration conditions and the effect of vibration on SMFL detection were analyzed. Finally, a characterization index for cable structure corrosion was proposed based on the theoretical model and experimental results. The main contributions of this paper concerning the published literatures are summarized as follows:(1)In terms of theory, the influence of vibration was analyzed based on the existing two-dimensional magnetic dipole model for rectangular corroded defects. The two-dimensional magnetic dipole model under vibration conditions was derived, and a theoretical model for the additional magnetic field under vibration was established. The original theoretical model was optimized.(2)In terms of application, this study goes beyond the previous ideal experimental conditions by considering the cable’s vibration effects generated by the coupling action between the environmental and alternating loads. Through empirical analysis, the impact of this vibration effect on the detection of leakage magnetic flux was assessed. This assessment lays the foundation for the accurate diagnosis of cable corrosion and wire breakage under varying environmental and load conditions.(3)In terms of diagnostic methods, an evaluation index *A* for cable corrosion under vibration conditions was proposed. This index effectively reduces the influence of vibration and improves the accuracy of cable corrosion diagnosis.

### 1.4. Organization of the Paper

The rest of this paper is organized as follows. In Section 2, the principle of spontaneous leakage magnetic flux detection is introduced, and optimizations are made based on the original two-dimensional magnetic dipole model. In Section 3, experimental tests on the spontaneous leakage magnetic flux detection of corroded steel strands under vibration conditions are conducted. In Section 4, the impact of vibration on the corrosion-induced leakage magnetic flux signals is analyzed from multiple perspectives. The evaluation index suitable for assessing the corrosion of steel strands under vibration conditions are proposed. The last section outlines some main conclusions.

## 2. Theoretical Background

According to the principle of SMFL detection, a galvanized steel strand, serving as a ferromagnetic material, produces a magnetic leakage field on its corroded surface. This field can be understood as the vector sum of several types of magnetic fields [46], as illustrated in Figure 1. When the cable is undamaged and stationary, the magnetic field measured by the magnetic detection instrument is only the vector sum of the environmental magnetic field ***B***_E_ and the self-induced magnetic field ***B***_I_. Hence, the measured magnetic induction intensity ***B***_S_ is expressed as Equation (1). However, when the cable experiences radial vibrations, the displacement variation leads to an increase in the magnetic field of the measured signal, Consequently, the magnetic induction intensity measured during vibration can be expressed as ***B***_V_, as shown in Equation (2). Once a corrosion defect occurs within the cable structure, a corrosion SMFL emerges near the defect and continuously changes under the influence of vibrations. The magnetic induction strength at rest, ***B***_S-C,_ can be defined as presented in Equation (1), whereas the magnetic induction strength during vibration, ***B***_V-C_, is expressed in Equation (2).
(1)$$B_{S}=B_{E}+B_{I}\to B_{S-C}^{}=B_{S}+B_{C}(c)$$
(2)$$B_{V}=B_{S}+\Delta B_{V}(0)\to B_{V-C}=B_{S}+\Delta B_{V}(c)+B_{C}(c)$$
where ***B***_E_ is the environmental magnetic field. ***B***_I_ is the structural self-induced magnetic field. ***B***_C_(*c*) is the corrosion self-magnetic flux leakage. ***B***_S_, ***B***_S-C_ are the magnetic induction at rest. ***B***_V_, ***B***_V-C_ are the magnetic induction during vibration, corresponding to the intact structure and the corroded structure. Δ***B***_V_(0), Δ***B***_V_(*c*) are the magnetic field increments affected by the vibration effect (*c* is the corrosion ratio).

Figure 1 shows the different degrees of displacement and defect shape changes during the vibration of the steel strand, which will cause the change in magnetic induction intensity. Δ***B***_V_(0) and Δ***B***_V_(*c*) are the increments of the magnetic field under the influence of vibration. To analyze the effect of vibration on the magnetic field and its varying behavior, two types of magnetic field increments are defined: One type is the magnetic field increment caused by displacement change, denoted as Δ***B***^dis^ and described by Equation (3). This increment is termed the displacement-added magnetic field. The other type is the magnetic field increment caused by defect deformation, represented as Δ***B***^def^ and outlined in Equation (4). This increment is referred to as the deformation-added magnetic field.
(3)$$\Delta B^{\mathrm{dis}}=B_{V}-B_{S}=\Delta B_{V}(0)$$
(4)$$\Delta B^{\mathrm{def}}=B_{V-C}-B_{S-C}-\Delta B^{\mathrm{dis}}=\Delta B_{V}(c)-\Delta B^{\mathrm{dis}}$$

### 2.1. Displacement-Added Magnetic Field

The steel strand vibrating under external load excitation can be regarded as an Euler beam, which only considers the bending deformation and ignores the effects of shear deformation and moment of inertia. Assuming the displacement function *u*(*x*,*t*) of the strand vibration response is known, Point P along the axial position *x*_1_ experiences an external magnetic induction ***B***_p_. A two-dimensional overall coordinate system, *xoz*, is established in this plane. Additionally, a local coordinate system, *x*_1_*oz*_1_, is established, where the *x*_1_ axial aligns with the axial direction of the strand, and the *z*_1_ axis is perpendicular to it, as shown in Figure 2.

The angle between the local coordinate system and the overall coordinate system is denoted as *θ*. Consequently, ***B***_p_ can be expressed as the vector sum of the two magnetic fields in the x-axis and z-axis directions. Assuming that the magnetic field component ***B***_p*x*_ in the *x*-direction follows a specific functional relationship ***B***_p*x*_
*= f*_m_(*x*), and the magnetic field component ***B***_p*z*_ in the *z*-direction satisfies another specific functional relationship ***B***_p*z*_
*= f*_n_(*y*), the external magnetic induction ***B***_P_ at the point P(*x*_1_,0) in the local coordinate system *x*_1_*oz*_1_ is described by Equation (5). During the vibration of the steel strand, point P moves to P’ at time *t*, where the location of P’ becomes (*x*_1_, *u* (*x*_1_, *t*)). The magnetic induction strength at P’, denoted as ***B***_P’,_ can be determined as Equation (6).
(5)$$B_{p}=\left[\begin{array}{c} f_{m}\left(x_{1}\times \cos\theta \right)\\ f_{n}\left(x_{1}\times \sin\theta \right) \end{array}\right]$$
(6)$$B_{p\prime }=\left[\begin{array}{c} f_{m}\left(x_{1}\cdot \cos\theta -u\left(x_{1},t\right)\cdot \sin\theta \right)\\ f_{n}\left(x_{1}\cdot \sin\theta +u\left(x_{1},t\right)\cdot \cos\theta \right) \end{array}\right]$$
(7)$$\Delta B^{\mathrm{dis}}=B_{p\prime }-B_{p}=\left[\begin{array}{c} \Delta B_{x}^{\mathrm{dis}}\\ \Delta B_{z}^{\mathrm{dis}} \end{array}\right]=\left[\begin{array}{c} f_{m}\left(x_{1}\cdot \cos\theta -u\left(x_{1},t\right)\cdot \sin\theta \right)\\ f_{n}\left(x_{1}\cdot \sin\theta +u\left(x_{1},t\right)\cdot \cos\theta \right) \end{array}\right]-\left[\begin{array}{c} f_{m}\left(x_{1}\cdot \cos\theta \right)\\ f_{n}\left(x_{1}\cdot \sin\theta \right) \end{array}\right]$$

The displacement-added magnetic field Δ***B***^dis^ can be approximated by Equation (7). It can be seen that the increment of the magnetic induction intensity Δ***B***^dis^ caused by the change of the vibration displacement of the steel strand is mainly related to the magnetic field strength around the steel strand and the vibration response displacement function. Combined with the knowledge of structural dynamics, it is known that the vibration response displacement function *u* (*x*, *t*) of the steel strand is a periodic function. Therefore, in a relatively stable geomagnetic field, the displacement-added magnetic field will fluctuate due to the periodicity of the displacement function.

### 2.2. Deformation-Added Magnetic Field

Δ***B***^def^ reveals the defective deformation effect of steel strands during vibration, and the characteristic law of Δ***B***^def^ under the vibration state is analyzed based on the magnetic dipole model. Suppose there is a rectangular defect with a width of 2*a* and a depth of *h*_0_ at position *x*_0_ of the strand at rest, as shown in Figure 3, then a right-angle coordinate system, *x*o*z*, is established at the center of the defect surface. On both sides of the defect, positive and negative magnetic charges are uniformly distributed. At a particular point Q in space, the magnetic field magnitudes ***H***_A_ and ***H***_B_ generated by all the magnetic charge microelements on surfaces A and B can be expressed as Equations (8) and (9). According to the magnetic field superposition principle, the magnetic induction intensity at point Q can be expressed as Equation (10).
(8)$$H_{A}=\left(\begin{array}{c} H_{Ax}\\ H_{Az} \end{array}\right)=\frac{M(\sigma ,H_{1}^{0})}{2\pi }\left(\begin{array}{c} \arctan\frac{-h}{x+a}+\arctan\frac{h+h_{0}}{x+a}\\ \frac{1}{2}\ln\frac{{\left(x+a\right)}^{2}+{\left(h+h_{0}\right)}^{2}}{{\left(x+a\right)}^{2}+h^{2}} \end{array}\right)$$
(9)$$H_{B}=\left(\begin{array}{c} H_{Bx}\\ H_{Bz} \end{array}\right)=\frac{M(\sigma ,H_{1}^{0})}{2\pi }\left(\begin{array}{c} \arctan\frac{-h}{-x+a}+\arctan\frac{h+h_{0}}{-x+a}\\ \frac{1}{2}\ln\frac{{\left(x-a\right)}^{2}+h^{2}}{{\left(x-a\right)}^{2}+{\left(h+h_{0}\right)}^{2}} \end{array}\right)$$
(10)$$B_{Q}=\mu_{0}\mu_{r}H_{Q}=\mu_{0}\mu_{r}\left(\begin{array}{c} H_{Qx}\\ H_{Qz} \end{array}\right)=\mu_{0}\mu_{r}\left(\begin{array}{c} H_{Ax}+H_{Bx}\\ H_{\mathrm{Az}}+H_{Bz} \end{array}\right)$$

When the strand vibrates at a certain moment t, the depth of the rectangular defect remains constant, while the corrosion width expands to 2*a*’, as depicted in Figure 3. The defect cross-section still remains perpendicular to the central axis and satisfies the assumption of flat cross-section. The coordinate transformation of the magnetic charge unit of A’ and B’ during the vibration process is obtained by Equation (11). Their respective magnetic field strengths ***H***_A’_ and ***H***_B’_ generated at point Q can be calculated by Equation (12). The total magnetic induction strength ***B***_Q’_ at point Q is calculated by integrating Equation (13).

R is the radius of the strand. M_0_ is the bending moment at the damage. EI is the bending stiffness of the section at the damage. k is the curvature of the central axis. *μ*_0_ and *μ*_r_ are the vacuum permeability and relative permeability of the material, respectively.
(11)$$\begin{array}{c} A\prime (x_{1},z_{1})\Rightarrow A\prime \left(-a-\frac{aM_{0}(R+z_{1})}{\mathrm{EI}},z_{1}\right)\Rightarrow A\prime \left(a(R+z_{1})\frac{\partial^{2}u(x_{0},t)}{\partial x^{2}}-a,z_{1}\right)\\ B\prime (x_{2},z_{2})\Rightarrow B\prime \left(a+\frac{aM_{0}(R+z_{2})}{\mathrm{EI}},z_{2}\right)\Rightarrow B\prime \left(a-a(R+z_{2})\frac{\partial^{2}u(x_{0},t)}{\partial x^{2}},z_{2}\right) \end{array}$$
(12)$$\begin{array}{l} H_{A\prime }=\left(\begin{array}{c} H_{A\prime x}\\ H_{A\prime z} \end{array}\right)=\int \frac{Mds}{2\pi {\left(r_{1}\prime \right)}^{2}}\vec{r_{1}\prime }=\frac{M}{2\pi }\int_{-h_{0}}^{0}\frac{\sqrt{1+{\left(\frac{1}{ak}\right)}^{2}}dy\left(\begin{array}{c} x+a-ak(R+z_{1})\\ h-z_{1} \end{array}\right)}{{\left[x+a-ak(R+z_{1})\right]}^{2}+{\left(h-z_{1}\right)}^{2}}\\ H_{B\prime }=\left(\begin{array}{c} H_{B\prime x}\\ H_{B\prime z} \end{array}\right)=\int \frac{Mds}{2\pi {\left(r_{2}\prime \right)}^{2}}\vec{r_{2}\prime }=\frac{M}{2\pi }\int_{-h_{0}}^{0}\frac{\sqrt{1+{\left(\frac{1}{ak}\right)}^{2}}dy\left(\begin{array}{c} x-a+ak(R+z_{2})\\ h-z_{2} \end{array}\right)}{{\left[x-a+ak(R+z_{2})\right]}^{2}+{\left(h-z_{2}\right)}^{2}} \end{array}$$
(13)$$B_{Q\prime }=\mu_{0}\mu_{r}H_{Q\prime }=\mu_{0}\mu_{r}\left(\begin{array}{c} H_{Q\prime x}\\ H_{Q\prime z} \end{array}\right)=\mu_{0}\mu_{r}\left(\begin{array}{c} H_{A\prime x}+H_{B\prime x}\\ H_{A\prime z}+H_{B\prime z} \end{array}\right)$$
(14)$$\Delta B^{\mathrm{def}}=B_{Q\prime }-B_{Q}=\mu_{0}\mu_{r}\left(\begin{array}{c} H_{Q\prime x}-H_{Qx}\\ H_{Q\prime z}-H_{Qz} \end{array}\right)=\left(\begin{array}{c} \Delta B_{x}^{\mathrm{def}}\\ \Delta B_{z}^{\mathrm{def}} \end{array}\right)$$

The difference between the measured magnetic induction intensity of the Q-point in the vibrating and stationary states is defined as the deformation-induced additional magnetic field Δ***B***^def^, see Equation (14). Equations (8)–(14) demonstrate that the self-leakage magnetic field generated by corroded defects is mainly related to the measurement distance *h*, defect depth *h*_0_, and defect width *a*. To further analyze the influence of *h*, *h*_0_, and *a* on the leakage magnetic field of defects, MATLAB 2021a software was used for numerical simulation of the defect leakage magnetic field ***B***_Q’_ in the vibrating state. The results of these simulations are presented in Figure 4.

Figure 4 demonstrates that both components of ***B***_Q’*x*_ and ***B***_Q’*z*_ possess symmetry, and the extremum and zero-crossing phenomena in the curve conform to the general distribution law of the SMFL method. Figure 4a–d indicate that the magnetic field intensity is positively correlated with defect depth *h*_0_ and defect width *a*. Figure 4e,f indicate that the magnetic field intensity is negatively correlated with the measurement distance *h*. *h*, *h*_0_, and *a* do not alter the distribution pattern of the magnetic field, but only affect the extremum. In fact, different degrees of *h*_0_ and *a* represent varying section loss rates of the steel strand, indicating different levels of corrosion. Therefore, in the subsequent experimental study, the degree of corrosion in the steel strand (denoted as c) is used to represent the differences in defect width *a* and depth *h*_0_. This theoretical model also serves as a guiding principle for the subsequent experimental research.

In summary, the alterations in magnetic field distribution within the strand during vibration can be categorized into two main aspects. First, the displacement of the strand leads to incremental magnetic induction intensity (Δ***B***^dis^) at the measurement point, which is primarily influenced by the stability of the surrounding magnetic field. Under relatively stable geomagnetic conditions, Δ***B***^dis^ will exhibit fluctuating changes. Second, the defect size undergoes varying degrees of change due to the vibration, resulting in an increase in corrosion self-magnetic flux leakage. This increase, denoted as Δ***B***^def^, is primarily determined by the defect’s size. The above simplified theoretical model effectively explains the impact of vibration on SMFL and provides the theoretical foundation for the following tests. Moreover, the model highlights the significance of both the stability of the surrounding magnetic field and the size of the defect in defining the incremental changes observed in the magnetic field distribution during vibration.

## 3. Methodology and Experiment

The experimental objective is to investigate the characteristics of SMFL in corroded steel strands under vibration and examine the effects of vibration on SMFL detection. As shown in Figure 5, experiments were conducted for tension, corrosion, vibration, and signal acquisition. The signals were collected subsequent to the corrosion process.

### 3.1. Preparation and Tension

The experiment material was a smooth steel strand (1 × 7Φ5). The strand had a nominal diameter of 15.20 mm and a single length of 7 m. Table 1 provides the chemical components and material properties of steel strand specimens. The relative permeability of steel strand *μ* = 2000, and the saturation magnetization intensity *M_s_* = 1 × 10^6^ A/m.

Four groups of experiments were performed in Table 2; 1# specimen was conducted to explore the effect of measuring distance on SMFL; 2# and 3# specimens were conducted to investigate the effect of corrosion degree on SMFL under different excitation conditions; and 4 # specimen was not corroded, and the purpose is to explore the effect of the change in displacement of the strand on the magnetic measurement in the vibration state.

To simulate real-life bridge conditions, the strand was intentionally corroded after being subjected to a tensioning force. Given that the length of the tensioning platform is 6 m, each strand was set to a length of 7 m to ensure an ample tensioning area. The team’s tensioning platform, demonstrated in Figure 5 as the “Tension area”, was primarily responsible for the tensioning work. The tensioning force is set to 70 kN.

### 3.2. Specimen Corrosion

Due to the long length and prior tensioning of the strand specimen, weighing the specimen itself was inconvenient. Therefore, in this experiment, the degree of corrosion of the steel strands was controlled using electrochemical corrosion and the empirical formula for the theoretical corrosion rate, denoted as *c.* Presented as the “Corrosion area” in Figure 5, the specimen to be corroded was placed in 5% NaCl electrolyte solution. A water-absorbent cloth was placed at the position to be corroded and immersed in the electrolyte solution. Moreover, the positive pole of an external power supply was connected at the specimen end. A carbon rod was placed in the solution to connect the negative pole of the power supply to form a complete electrochemical corrosion closed circuit.

By applying Equation (15) and setting the constant DC power supply’s current *I* = 0.4 A, the corrosion degree *c* of the specimen could be determined. The actual corrosion effect of the steel strand specimen is shown in Figure 6. It is found that with the increase of corrosion time, the area of the corrosion section of the steel strand was significantly reduced.
(15)$$c=\frac{\Delta m_{\mathrm{Fe}}}{m_{}}=\frac{M_{\mathrm{Fe}}It}{nF\lambda l}=2.63\times 10^{-7}\frac{It}{l}$$
where *I* is the current size, *t* is the corrosion time, *l* is the strand corrosion width. M_Fe_ is the atomic molar mass of Fe, M_Fe_ = 55.85 × 10^−3^ kg/mol. *n* is the transfer of electrons in the anodic reaction, *n* = 2. F is the Faraday constant, F = 96,485 C/mol. λ is the theoretical weight of the strand per meter, λ = 1.101 kg/m.

### 3.3. Specimen Vibration and Magnetic Signal Acquisition

A modal exciter (Chongqing, China, Sa-jz005) was placed at the middle of the specimen to excite the steel strand to simulate the cable vibration generated by external loading during service. A low-frequency signal generator was used to simulate the sinusoidal signal of the cable vibration. A power amplifier drove the exciter. The modal exciter was a permanent magnet electric modal one. When the power amplifier supplied variable-frequency current to the moving coil, the exciting force is shown in Equation (16).
(16)F=BIL
where *B* is the average magnetic induction strength, *I* is the instantaneous value of the current supplied by the power amplifier, *L* is the effective length of the coil wire. Therefore, the output frequency of the signal generator controlled the vibration speed, whereas the output current of the power amplifier controlled the exciting force.

In order to distinguish the magnetic leakage signal before and after the vibration of the steel strand, three types of magnetic signal were recorded in the experiments: initial magnetic signal ***B***_S_ (without corrosion), stationary magnetic signal ***B***_S-C_ (The corrosion rate is *c*), and vibrating magnetic signal ***B***_V-C_ (The corrosion rate is *c*). When the corrosion degree *c* = 0, the vibrating magnetic signals ***B***_V-C_ (0) = ***B***_V_.

The magnetoresistive sensor (HMR 2300) was used to obtain the magnetic field distribution of the specimens. The resolution of the sensor was 67μGs with 10–157 sampling points per second, and the output ranged from −2Gs–2Gs. In the magnetic field measurement system, the magnetic sensor was carried by the positioning device to move freely on the structure’s surface. The measurement area has a length of 40 cm, divided into 41 measurement points, each point is 1 cm away from the other, as shown in Figure 7. The magnetic sensor has three directional components (abbreviated as ***B****_x_*, ***B****_y_*, and ***B****_z_*) and only the tangential (***B****_x_*) and normal (***B****_z_*) components are needed for the measurement. The lift-off height *d* is the distance between the magnetic sensor and the surface of the specimen and the measurement heights of this experiment are 1 cm, 2 cm and 3 cm. During the vibration, the magnetic field data were collected three times at each data point. The Scanning scheme of magnetic field signals is shown in Figure 7.

## 4. Results and Discussion

The experiment results are analyzed to investigate the difference in the SMFL of the corroded steel strand between vibration and static states, while the interference of the external magnetic and geomagnetic fields needed to be eliminated. To achieve this, Equation (17) is applied to eliminate the effects caused by the environmental magnetic field and the structural self-induced magnetic field. Subsequently, 1# specimen (corrosion ratio *c* = 8%) is taken as an example and its distribution laws of the corrosion SMFL are shown in Figure 8.
(17)$$B_{C}=B_{S-C}-B_{S}$$

It can be found from Figure 8 that the tangential component curve has a minimal value in the center of the corrosion region. On the other hand, the normal component curve crosses the zero point and has two extreme values in the corrosion region. There is a notable correlation between the extremum points and the corrosion area, which aligns with the outcomes obtained from the magnetic dipole model [26].

### 4.1. SMFL in Vibration and Stationary States

In order to investigate the distribution characteristics and evolution law of cable structures’ self-magnetic flux leakage (simplified as “corrosion SMFL”) under vibration and stationary states, the effects caused by the environmental magnetic field and the structural self-induced magnetic field are eliminated by Equation (17). The following paper analyzes the characteristics of the corroded SMFL under vibration and stationary states from two aspects: different lift-off heights and corrosion degrees. Where, ***B***_C_ and **$B_{C}^{V}$** corresponded to the stationary state and vibration state of cable structure. ***B***_C*x*_ and ***B***_C*z*_ corresponded to the tangential and normal components of the corrosion SMFL curve in stationary state. $B_{Cx}^{V}$ and $B_{Cz}^{V}$ corresponded to the tangential and normal components of the corrosion SMFL curve in vibration state.

#### 4.1.1. Different Lift-Off Heights

Considering the spatial nature of the self-magnetic flux leakage (SMFL) field generated by the defect, it is essential to note that the signal strength of this field is affected by the sensing locations. The theoretical analysis indicates that the SMFL of corrosion damage obtained by different lift-off heights of the testing equipment is different. Therefore, the 1# specimen is taken as an illustration. The distribution curves of SMFL field under the same corrosion condition but in different measuring lift-off heights are analyzed and shown in Figure 9.

As shown in Figure 9, on the one hand, the corrosion SMFL in the vibration state has similar characteristics to the stationary state. The SMFL curves for various lift-off heights all demonstrate signal extremes near the corrosion area (highlighted by the gray line box). In the ***B***_C*x*_ and $B_{Cx}^{V}$ curves, each signal curve under different lift-off heights displays consistent behavior, where the signal strength of the magnetic field changed reciprocally to the distance. Specifically, The higher the lift-off height, the lower the signal strength. In the ***B***_C*z*_ and $B_{Cz}^{V}$ curves, the magnetic signals are also inversely proportional to the lift-off heights. Two extreme points emerge on each side of the corrosion area, while one intersection point is identified at the center of the corrosion area. The experimental results are consistent with the numerical simulation results presented in the previous theoretical section.

On the other hand, although the corrosion SMFL in the vibration state has similar characteristics to the stationary state, there are still varying degrees of fluctuations which are particularly significant at the curve’s inflection point. With the increase of the measurement lift-off heights, the ***B***_C_ and $B_{C}^{V}$ curves tend to be flatter, resulting in a smaller extreme value near the corrosion area. This feature indicates that the influence of the measurement lift-off heights on the SMFL curve is similar for both vibration and stationary states. To mitigate the impact of the measurement lift-off heights on the SMFL detection, the sensor should be as close as possible to the measured part during the detection and choose the appropriate measurement distance.

#### 4.1.2. Different Corrosion Degrees

To explore the characteristics of the SMFL in different corrosion degrees of the strand under vibration and stationary states, the 2# specimen is taken as an example and the results of the analysis are shown in Figure 10. ***B***_Cx_ and ***B***_Cz_ corresponded to the tangential and normal components of the corrosion SMFL curve.

From Figure 10, it is evident that the corrosion SMFL signals in the vibration state fluctuate around the stationary SMFL curve and follow a similar trend with it under different corrosion degrees. The changes in the SMFL curves under different corrosion degree are obvious. As the corrosion degree increases, the peak of the tangential and normal components of the corrosion SMFL curve increases significantly, indicating that the extreme value of the curve is positively correlated with the corrosion degree. The normal component ***B***_Cz_ and $B_{Cz}^{V}$ curves display double polar points and cross the zero threshold, while the $B_{Cz}^{V}$ signals have an overall upward shift based on the ***B***_Cz_ curve. In the ***B***_Cx_ and $B_{Cx}^{V}$ curves, the SMFL signals with different corrosion degrees have obvious intersection points near the corrosion area.

The width Δ*x* of the two intersection points is greater than the width of the corrosion area, which is about twice the corrosion width [33]. Through the location and width of Δ*x*, it can be qualitatively determined regarding the location and width of the corrosion area. The extreme value point is positively correlated with the degree of corrosion, which can be used to characterize the degree of corrosion. The relationship between the degree of corrosion of steel wire ropes and their self-induced magnetic leakage signals also agrees with the results of numerical simulation.

In summary, the extreme point of the SMFL curve is negatively correlated with the measured lift-off height and positively correlated with the corrosion degree. The corrosion SMFL signals in the vibration state fluctuate around the stationary SMFL curve and have the same trend and characteristics. In order to analyze the fluctuation characteristics of the SMFL signals in the vibration state and to explore the impact of vibration on the quantification and localization of the corrosion area, the two aspects will be discussed later: vibration displacement effect and vibration deformation effect.

### 4.2. Analysis of the Vibration Effects on SMFL

#### 4.2.1. Displacement Effects (Δ***B***^dis^)

Theoretical analysis demonstrates that the uncorroded strand is only affected by displacement change during vibration. Therefore, the 4# specimen is taken as an example and its displacement-added magnetic field Δ***B***^dis^ can be derived from Equation (3). Figure 11 presents the statistical characteristics of normal (Δ$B_{z}^{\mathrm{dis}}$) and tangential (Δ$B_{x}^{\mathrm{dis}}$) components of the Δ***B***^dis^ signals under different excitation currents, represented as a violin diagram. In this diagram, the white vertical line represents the magnetic signal range from 1/4 to 3/4. The yellow vertical line represents the average value and the area of the violin represents the distribution interval and size of this set of data. It can be seen from Figure 11 that the curves of the Δ***B***^dis^ signals fluctuate up and down with no apparent change pattern and 50% of the magnetic signals are concentrated within −1–1 mGs. The overall signals exhibit a “high in the middle and low on both sides” pattern, remaining within the ±5 mGs range. This feature indicates that the fluctuation of the Δ***B***^dis^ signals is slight because the external magnetic induction intensity around the steel strand in the experiment chamber is relatively stable. Therefore, the variation in the Δ***B***^dis^ signals is mainly affected by the vibration amplitude. The displacement-added magnetic field only fluctuates up and down in a small range when the vibration amplitude of the steel strand is not significant.

Figure 11 reflects that the form of Δ$B_{x}^{\mathrm{dis}}$
*x* is “narrow and high” and the shape of Δ$B_{z}^{\mathrm{dis}}$ is “wide and flat”, indicating that the normal component of the Δ$B^{\mathrm{dis}}$ signals is more discrete than the tangential component. To assess the fluctuation of the data set, the variance *s*^2^ of 4# specimen is calculated by Equation (18). The results are depicted in Figure 12. Figure 12a–c represent the variance results of the excitation frequency of 10 Hz, 8 Hz, 5 Hz. It can be shown from Figure 12 that the variances of Δ$B_{z}^{\mathrm{dis}}$ are approximately 0.2 mGs^2^ greater than those of Δ$B_{x}^{\mathrm{dis}}$, indicating that the degree of fluctuation of the normal component is greater than that of the tangential component. This phenomenon can be attributed to the experimental setup, which applies radial vibration to the strand in the same direction as the normal component of the magnetic signals. Consequently, the normal displacement of the strand is larger than the tangential displacement. The results show that the strand vibration has a more pronounced impact on the normal component of the magnetic signal.
(18)$$s^{2}=\frac{\sum_{i=1}^{n}{\left(x_{i}-\bar{x}\right)}^{2}}{n}$$
where *x_i_* is the value of tangential and normal components of Δ***B***^dis^, $\bar{x}$ is the mean of the set of data, *n* is the quantity of data in the group.

Considering the fluctuation of Δ***B***^dis^, Figure 13 analyzes the statistical distribution law of Δ***B***^dis^ under other excitation conditions in the form of a histogram. Where β denotes the range of Δ***B***^dis^, each data group is divided into intervals of 0.1 mGs and F is the frequency of each interval. Similar to Figure 11, the overall magnetic signals are “high in the middle and low on both sides”, mainly distributed around the zero value. The statistical distribution of the data at different excitation frequencies has no obvious variation pattern, but all show characteristics of normal distribution. The maximum frequency corresponds to the data distribution interval β_max_ around “0”, which indicates that the data’s fluctuation range is small.

When the external magnetic induction intensity around the strand is stable, the magnitude of the displacement-added magnetic field is mainly influenced by the vibration amplitude. The Δ***B***^dis^ signals fluctuate up and down around the zero value with the statistical characteristics of a normal distribution. The effects of displacement on SMFL are small, but the normal component fluctuated more. Therefore, the subsequent analysis would be carried out only for the tangential component.

#### 4.2.2. Corrosion Defect Deformation Effects (*Δ**B**^def^*)

Figure 1 shows that the corrosion SMFL is generated when the strand is corroded. It is shown from Figure 3 that the defect width changes when the strand vibrates, which affects the magnitude of the SMFL. To investigate the influence law of defect deformation on the SMFL during vibration, the 2# and 3# specimens are taken as examples. The analysis is carried out only for the tangential component of SMFL because the stability of the tangential component is better. Figure 14a,c show the SMFL at different corrosion degrees, where $B_{Cx}^{V}$ and ***B***_C*x*_ corresponded to the vibration and stationary states of the strand. Figure 14b,d are the results of the vibrating magnetic signals minding the static magnetic signals, which included the Δ$B_{x}^{\mathrm{dis}}$, as shown in Equation (19). The absolute value of Δ***B***_V*x*_ poles is shown in Equation (20).
(19)$$\Delta B_{Vx}=B_{Cx}^{V}-B_{Cx}=\Delta B_{x}^{\mathrm{dis}}+\Delta B_{x}^{\mathrm{def}}$$
(20)$$\eta =\left|\Delta B_{Vx}(\min)\right|$$

From Figure 14a,c, it is evident that the SMFL curves of the strand exhibit a strong correlation with the corrosion degree. The vibration magnetic signals are consistent with the variation pattern of the stationary magnetic signal, fluctuating around the stationary SMFL curve. Figure 13b,d show that the Δ***B***_V*x*_ signals oscillate near the zero value in the uncorroded region spanning from 0–15 cm and from 25–40 cm. This indicates that the displacement changed mainly occur in the uncorroded region, consistent with the characteristics of the Δ***B***^dis^. In the vicinity of the corroded region, Δ***B***_V*x*_ displays a more prominent variation with obvious extreme values, indicating an increase in defect width during vibration. From the relationship graph of η-*c*, it is seen that the extreme value of Δ***B***_V*x*_ is larger as the corrosion degree increases and has a linear correlation. It means that the greater the degree of corrosion, the greater the effect of vibration on SMFL.

On the other hand, Figure 13b,d also reveal that when the degree of corrosion of the steel strand rope ranges between 2% and 10%, the peak value of Δ***B***_V*x*_ does not exceed 10 mGs, and in the non-corroded area, Δ***B***_V*x*_ fluctuates within the range of −5 to 5 mGs. Δ***B***_V*x*_ represents the difference in magnetic signal between vibration and static station, indicating the influence of vibration. These experimental results demonstrate that the impact of vibration on magnetic flux leakage detection is correlated with the degree of corrosion. The greater the degree of corrosion, the greater the influence of vibration, primarily reflected in the fact that the deformation-added magnetic field is greater than the displacement-added magnetic field.

The relationship between η and *c* for the remaining excitation conditions of specimens 2# and 3# is illustrated in Figure 15a–e. There is an excellent linear relationship between η and c for different excitation conditions, demonstrating that η increased with the increase of corrosion degree *c*. Due to some specific errors in the experiments, certain working conditions do not confirm this linear relationship. The results of all conditions are fitted to Figure 15f. From the fitting result, the linear relationship between η and c is well fitted with a fitted regression rate R^2^ = 0.87. The corresponding linear equation is shown in Figure 15f. Considering that η represents the difference between vibration and rest at the extremes of the tangential component of the SMFL curve, it follows that this difference increases with the corrosion ratio. The extreme value of the SMFL curve is often used to characterize the corrosion degree of the strand, and the greater the corrosion degree, the greater the error in describing the corrosion in the vibration state.

In summary, the effect of vibration deformation on SMFL is more significant than that of vibration displacement and increases with the increase of corrosion degree. If the extreme point of the SMFL curve is used as the characteristic value to characterize the corrosion degree, there will be a large error. Therefore, it is necessary to reduce the influence of vibration on corrosion characterization as much as possible and to seek the characteristic value suitable for corrosion characterization under the vibration state.

### 4.3. Characterization Index of SMFL under Vibration

According to the analysis of the vibration effects on SMFL in Section 4.2, it is evident that the tangential component of SMFL exhibits higher stability. Consequently, the characterization index for corrosion is selected from the tangential component curve. As shown in Figure 16, M, N and P are the extreme value points on the SMFL curve. The experiment results show that points M, N are located on both sides of the corrosion area, with the width of the two intersection points (Δx) being approximately twice the width of the corrosion. Furthermore, the extreme value point P is positioned at the center of the corrosion area and exhibits a linear correlation with the degree of corrosion. Considering the vibration effects on the minima P of the SMFL curve, the three characteristic points (M, N, P) are used as cut-off points to analyze the characterization value of the corrosion degree.

In previous studies on the characterization model for corrosion in cables, only the magnetic signal values at points M, N, and P were used as reference points to construct indicators [20]. However, the theoretical and experimental research results mentioned earlier found that the magnetic signals between points M and N fluctuate to varying degrees due to vibration. Furthermore, as the degree of corrosion increases, the fluctuation at the extremum point P also becomes larger. Therefore, considering only the magnetic signals at these three reference points can lead to significant errors in determining the degree of corrosion. Therefore, this study incorporates the magnetic signals between points M and N into the construction of the characterization indicator. The average value is taken to reduce the influence of signal fluctuations, resulting in the construction of the characterization indicator *A* as expressed in Equation (21).

In Figure 16, ***B***_M_ = ***B****_x_*_1_, ***B***_N_ = ***B****_x_*_n_, Q is the midpoint of M and N,***B***_Q_ = 1/2(***B****_x_*_1_+ ***B****_x_*_n_). The magnitude of the magnetic signal at any point between M and N is ***B****_x_*_i_. The index *A* is constructed and its expression is presented in Equation (21).
(21)$$A=\frac{\sum_{i=1}^{n}B_{xi}-\frac{1}{2}(B_{x1}+B_{xn})}{n}$$

To verify the correlation between index *A* and corrosion degree *c*, the 2# and 3# specimens are taken as examples and the index *A* under different corrosion degrees is calculated by Equation (21). At the same time, the indicators mentioned in document [20] are also calculated using the vibration state magnetic signals of specimens 2# and 3# in this study. These calculated indicators are denoted as A’. The purpose is to compare whether the indicators A proposed in this paper are more accurate than those in previous research. The results are shown in Figure 17. Figure 17a,b are the calculation results of the characteristic index of specimen 3#. Figure 17c,d are the calculation results of the characteristic index of specimen 2#.

It is shown from Figure 17 that the relationship between the index *A* and the corrosion degree *c* can be approximately regarded as a linear relationship, satisfying the equation *A* = a + b ∗ c, where a is the slope of the fitted straight line and b is the intercept of the fitted straight line. To assess the potential of index A for corrosion detection in cable structures under vibration, the fitting effect of the equation is analyzed. This analysis includes both the significance test and the fitting test. The significance test of the regression equation is usually the *F* test. The principle is to decompose the total sum of squares of deviations into regression sums of squares and residual squares and see how many regression sums of squares and residual sums of squares are included in the total sum of squares of deviations. Its specific expression was determined by Equation (22).
(22)$$F=\sum_{i=1}^{n}(\hat{y_{i}}-\bar{y}{)}^{2}\times {\left(\frac{\sum_{i=1}^{n}(y_{i}-\hat{y_{i}}{)}^{2}}{n-2}\right)}^{-1}$$
where $\sum_{i=1}^{n}(\hat{y_{i}}-\bar{y}{)}^{2}$ is the regression sum of squares, $\sum_{i=1}^{n}(y_{i}-\hat{y_{i}}{)}^{2}$ is the residual sum of squares.

*F_α_*(1, n − 2) is the critical value, when *F* > *F_α_*, the regression equation is significant. To illustrate the fitting effect of the regression line, the goodness of fit *R*^2^ of the regression equation is further calculated. The results are shown in Table 3. Figure 17 and Table 3 show that the four groups of data all satisfied P < 0.05, so the regression equation is significant. In fact, for the univariate linear regression, the significance of the regression equation is equivalent to the test of the regression coefficient. *R*^2^ values of all samples are more significant than 0.95, indicating that the data in the working stage with cracks had a higher linear relationship.

Furthermore, the calculation results of *R*^2^ in Table 3 indicate that the calculated results of indicator A in specimens 2# and 3# exhibits a superior performance compared to A’. This indicates that indicator A possesses a stronger correlation with corrosion level. Indicator A can reduce the influence of vibrations to some extent. Therefore, using the characteristic parameter *A* for corrosion characterization is feasible and it can provide more accurate determination results.

## 5. Conclusions

Taking the corrosion of steel strand under vibration as the focus of this study, the change of magnetic leakage signal of corroded steel strand under vibration was analyzed theoretically and the concept of additional magnetic field was put forward. The existence of additional magnetic field was verified by experiments and its variation characteristics were further analyzed, which provided a new idea for cable corrosion detection in practical engineering. Based on the test results, the correlation between magnetic characteristic index and corrosion degree was analyzed, and the main conclusions were as follows:(1)The corrosion SMFL signal in the vibration state fluctuated around the stationary SMFL curve, exhibiting similar trends and characteristics. The extreme point of the SMFL curve was negatively correlated with the measured lift-off height and positively correlated with the corrosion degree.(2)The influence of vibration on magnetic field distribution of cable structure was defined as the displacement-added magnetic field (Δ***B***^dis^) and the deformation-added magnetic field (Δ***B***^def^). The tangential components of both have good data stability. Δ***B***^dis^ followed a statistical normal distribution, fluctuating around zero value, and had no obvious relationship with excitation current and frequency. The effect of Δ***B***^def^ was mainly related to the degree of corrosion. The larger the degree of corrosion, the greater the impact of Δ***B***^def^. When the external magnetic induction intensity was stable, the increment of magnetic field intensity change was mainly determined by Δ***B***^def^.(3)Combined with the characteristics of SMFL curve under vibration and the influence of Δ***B***^dis^ and Δ***B***^def^ on the SMFL curve, a corrosion index *A* was proposed. Moreover, through testing and analyzing, the index *A* exhibited a strong linear fit with the degree of corrosion (with *R*^2^ values greater than 0.97 in all cases) and can reduce the influence of vibrations to some extent. Using index *A* for corrosion characterization was feasible and can provide more accurate determination results.

In summary, considering the influence of vibration effect on magnetic leakage detection, this study proposed a suitable characterization index for cable corrosion detection under vibration. This approach effectively improved the accuracy of cable corrosion detection. However, this method is also affected by environmental magnetic field, magnetization difference of specimens and other factors and the influence of temperature on magnetic leakage detection was not taken into account. These questions would be the focus of future research.

## Figures and Tables

**Figure 1 sensors-23-07130-f001:**
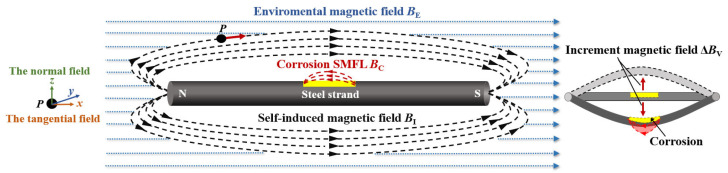
Magnetic field distribution of cable structure.

**Figure 2 sensors-23-07130-f002:**
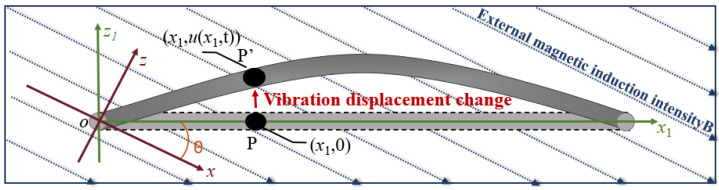
Schematic diagram of displacement-added magnetic field.

**Figure 3 sensors-23-07130-f003:**
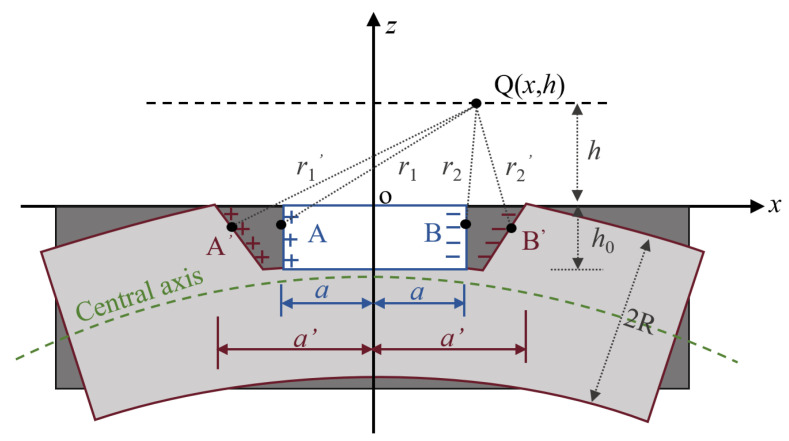
Schematic diagram of the shape change of the defect.

**Figure 4 sensors-23-07130-f004:**
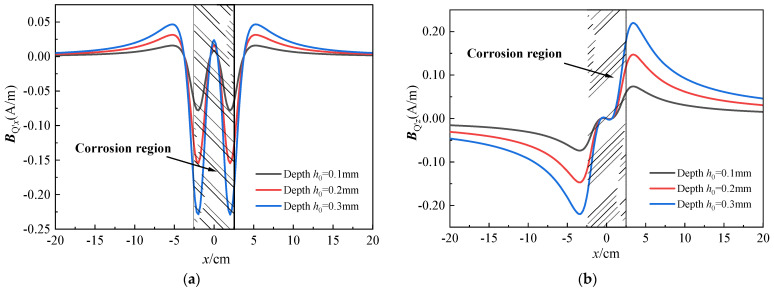
Theoretical distribution law of ***B***_Q’_: (**a**,**b**) are the x-axial component and z-axial component under various depths; (**c**,**d**) are the x-axial component and z-axial component under various widths; (**e**,**f**) are the x-axial component and z-axial component under various distances.

**Figure 5 sensors-23-07130-f005:**
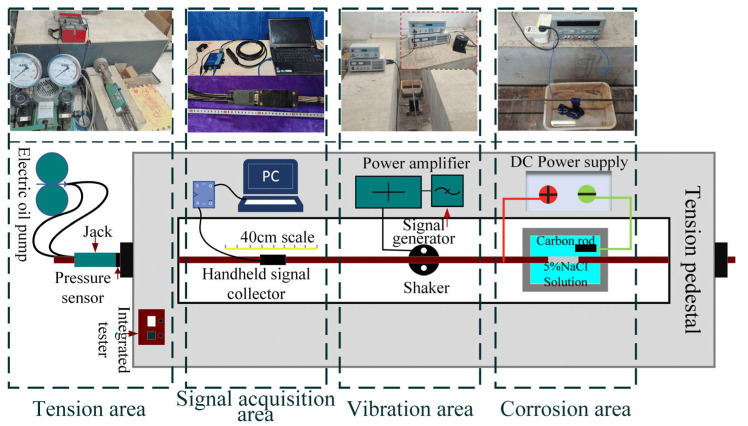
Schematic of the test work platform.

**Figure 6 sensors-23-07130-f006:**
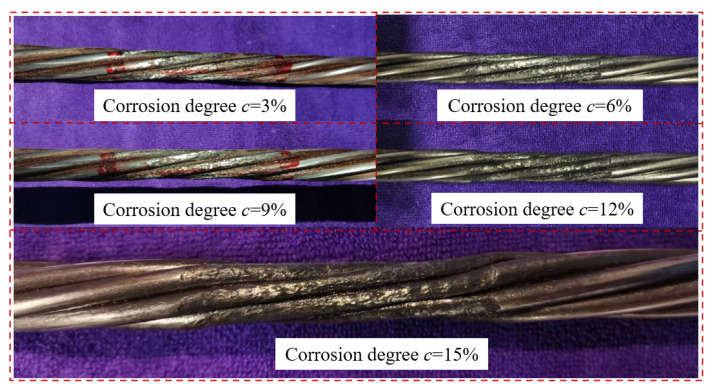
The actual corrosion effect of the specimen.

**Figure 7 sensors-23-07130-f007:**
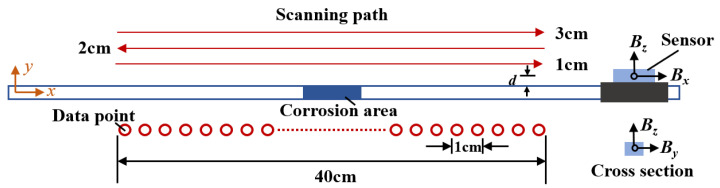
Scanning scheme of magnetic field signals.

**Figure 8 sensors-23-07130-f008:**
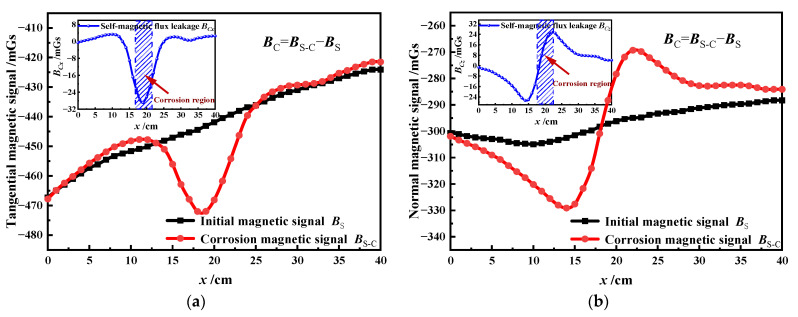
Calculation of the SMFL of corrosion damage: (**a**)tangential magnetic signal; (**b**)normal magnetic signal.

**Figure 9 sensors-23-07130-f009:**
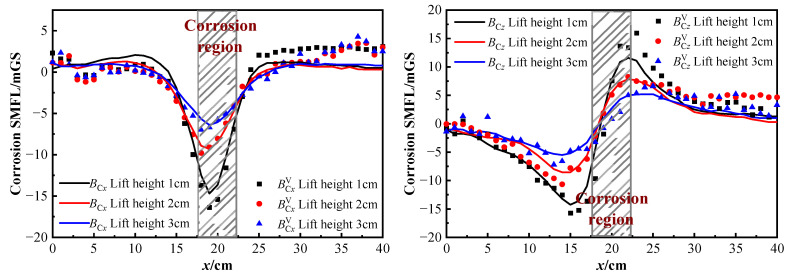
Corrosion SMFL with different lift-off heights.

**Figure 10 sensors-23-07130-f010:**
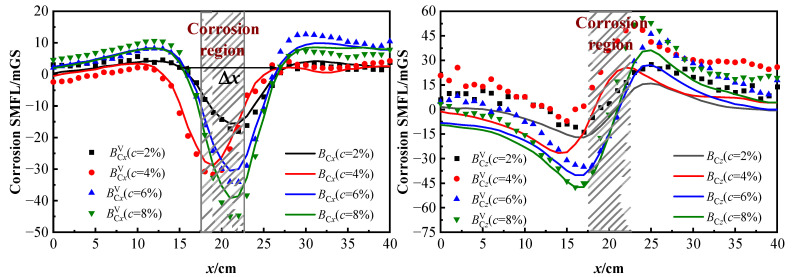
Corrosion SMFL with different corrosion degrees.

**Figure 11 sensors-23-07130-f011:**
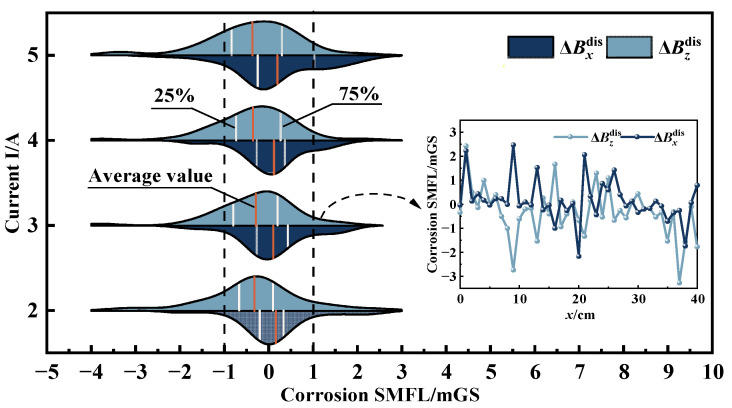
Displacement-added magnetic field of 4# specimen: f = 8 Hz.

**Figure 12 sensors-23-07130-f012:**
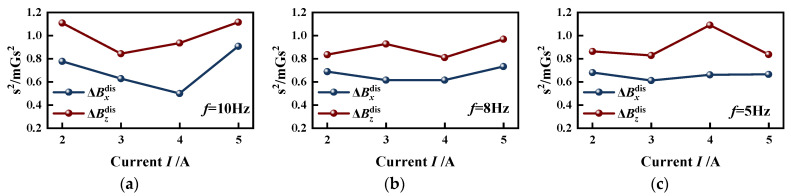
Variances of Δ$B_{x}^{dis}$ and Δ$B_{z}^{dis}$ of 4# specimen under different excitation conditions: (**a**) *f* = 10 Hz; (**b**) *f* = 8 Hz; (**c**) *f* = 5 Hz.

**Figure 13 sensors-23-07130-f013:**
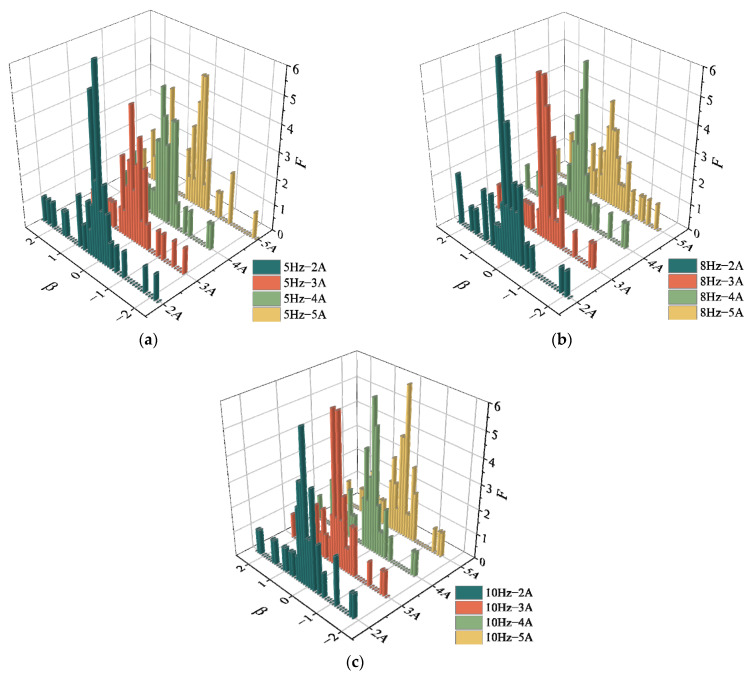
Statistical histogram of Δ***B***^dis^ of 4# specimen under different excitation conditions: (**a**) *f* = 10 Hz *I* = 2A–5A; (**b**) *f* = 8 Hz *I* = 2A–5A; (**c**) *f* = 5 Hz *I* = 2A–5A.

**Figure 14 sensors-23-07130-f014:**
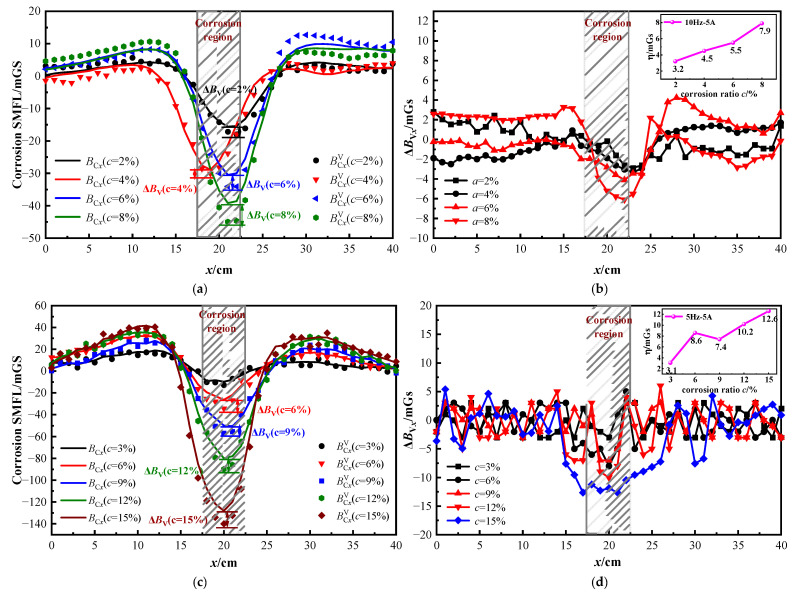
Corrosion SMFL and Δ***B***_Vx_ of 2# and 3# specimens under different corrosion degrees: (**a**) and (**b**) are 2# specimens; (**c**,**d**) are 3# specimens.

**Figure 15 sensors-23-07130-f015:**
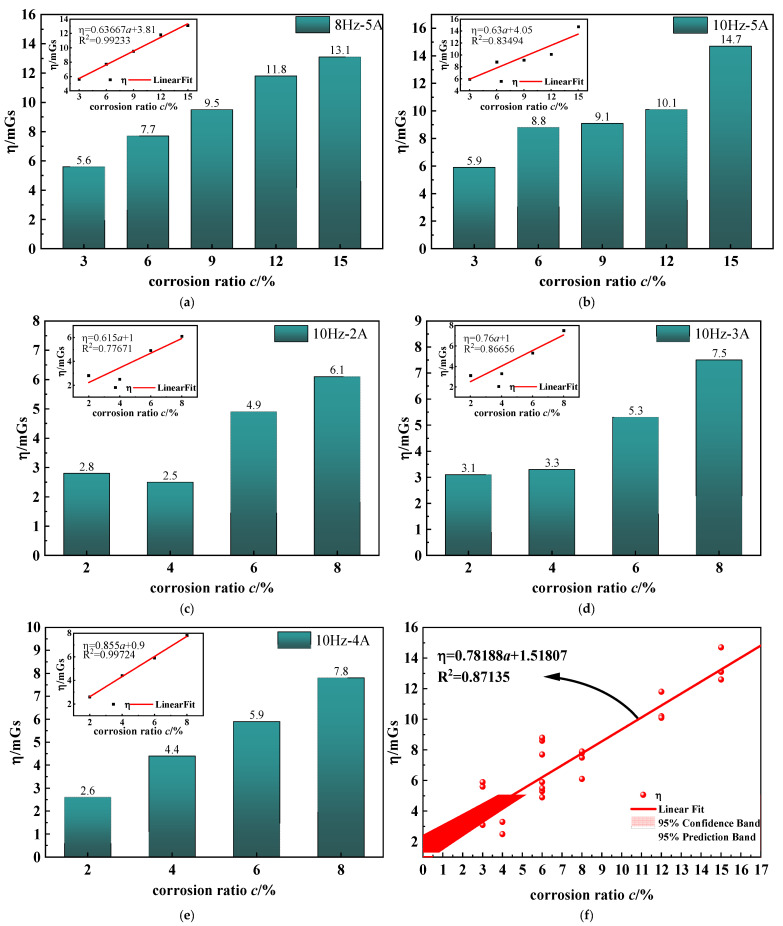
Relationship of η and c for the remaining excitation conditions of specimens 2# and 3#: (**a**,**b**) are 2# specimens; (**c**–**e**) are 3# specimens; (**f**) is the fitting result of η and c.

**Figure 16 sensors-23-07130-f016:**
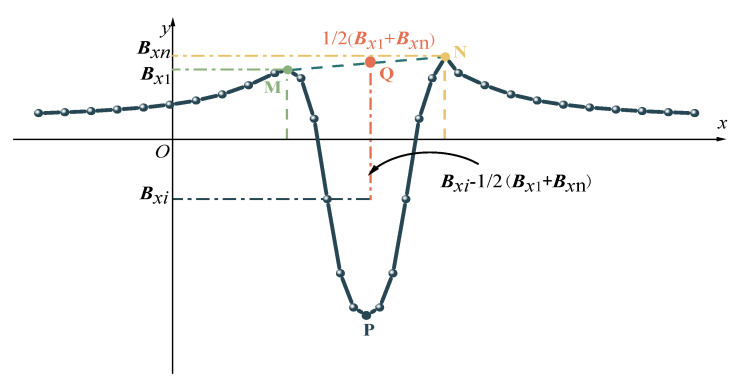
Schematic diagram of magnetic characteristics parameter.

**Figure 17 sensors-23-07130-f017:**
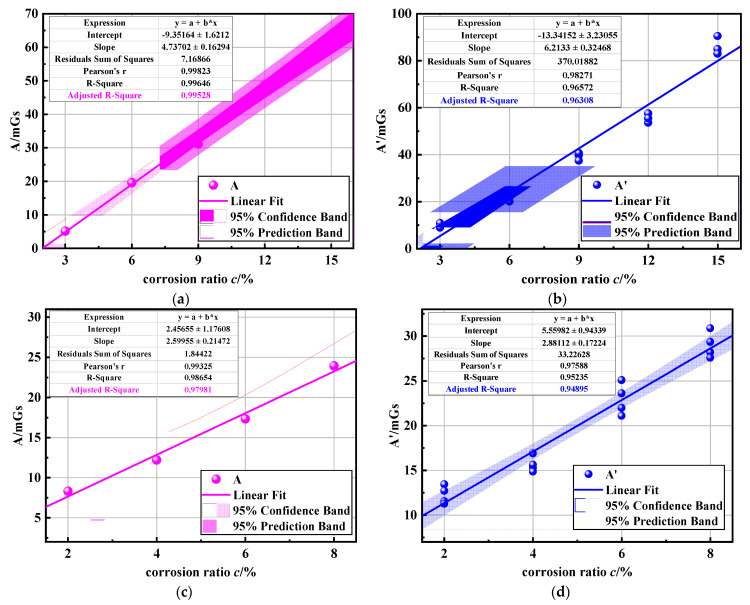
Correlation between parameters *A* and corrosion ratio *c*: (**a**,**b**) were 3# specimens; (**c**,**d**) were 2# specimens.

**Table 1 sensors-23-07130-t001:** Chemical components and material properties of steel strand specimens.

Material Proportion	Tensile Strength	Nominal Diameter	Elasticity Modulus	Weight	Elongation	Yield Load Fp0.2
Values	1860 MPa	15.20 mm	195 GPa	1.101 kg/m	≥3.5%	≥229 kN
Chemical compositions	C	Si	Mn	P	S	Cu
Proportion	0.8–0.85%	0.12–0.32%	0.6–0.9%	<0.025%	<0.025%	<0.2%

**Table 2 sensors-23-07130-t002:** Specimen grouping.

Number	Corrosion Loss Ratio *c*	Frequency *f*/Hz	Current *I*/A	Measuring Distance *d*/cm
1#	4%, 8%	5	10	1, 2, 3
2#	2%, 4%, 6%, 8%	10	2, 3, 4, 5	1
3#	3%, 6%, 9%, 12%, 15%	5, 8, 10	5	1
4#	0%	5, 8, 10	2, 3, 4, 5	1

**Table 3 sensors-23-07130-t003:** Linear fitting parameters.

Number	*F*	P	*R* ^2^
3#-A	854.155	0	0.99
3#-A’	366.208	0	0.96
2#-A	146.569	0.0067	0.98
2#-A’	664.068	0	0.95

## Data Availability

The data presented in this study are available from the first and corresponding author upon request. The data is not publicly available due to the policy of the data provider.

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
