# Peer review of "Experimental Analysis of the Magnetic Leakage Detection of a Corroded Steel Strand Due to Vibration"

_sensors, 2023, doi:10.3390/s23167130_

Round 1

Reviewer 1 Report

The present work addressed a very interesting and crucial topic in applying the magnetic signal to detect the corrosivity of steel strands under vibration. In general, the work is well designed and organised in a good logical flow, with a list of well-prepared figures and tables to support the conclusion. Before the acceptance, some minor revisions are suggested as below:

(1) Abstract: please upgrade to highlight: the background, state of the art, research gaps, aim of study, methodology, major finding & impact, etc.

(2) The authors assume rectangular defects in the second section of the article. However, this assumption seems to deviate from the realistic morphology of natural corroded defects, which are usually irregular in shape. Could the authors elucidate the rationale behind such a simplification and its implications for the accuracy and applicability of the results?

(3) It is well-known from engineering practice that the region adjacent to the cable anchor is prone to severe corrosion. How effective is the proposed SMFL approach for this part? Are there any limitations or challenges that need to be addressed?

(4) The work introduces a method for detecting corrosion defects using magnetic signals. However, the performance of this method may be affected by various environmental factors. Could the authors elaborate on what are the potential sources of interference or noise that can degrade the quality or reliability of the magnetic signals? How can these factors be mitigated or eliminated?

Although the English language is at an acceptable level in general, further polishment is highly suggested to improve the readability and coherence of the work.

Author Response

Responses to Reviewer #1 Comments

Dear Reviewer:

Firstly, we greatly appreciate your comments and suggestions on our manuscript. Those comments are valuable and helpful for revising and improving our paper and the essential guiding significance to our studies. We have studied all the comments and suggestions carefully. Then we have made revisions that can meet your approval. The responses to your comments are presented as follows:

Point 1:

Abstract: please upgrade to highlight: the background, state of the art, research gaps, aim of study, methodology, major finding & impact, etc.

Response 1:

Thanks for your comment. Our responses are as follows:

We have thoroughly restructured the research background, objectives, methods, main findings, and impacts discussed in the article. Additionally, we have made necessary adjustments and revisions to refine both the structure and content of the abstract. For specific details regarding these modifications, please refer to the abstract section of the manuscript, specifically lines 9~21.

Point 2:

The authors assume rectangular defects in the second section of the article. However, this assumption seems to deviate from the realistic morphology of natural corroded defects, which are usually irregular in shape. Could the authors elucidate the rationale behind such a simplification and its implications for the accuracy and applicability of the results?

Response 2:

Thanks for your comment. Our responses include two aspects:

(1) The corrosion investigated in this study pertains to localized rusting of ferromagnetic materials. It is acknowledged that there are discrepancies between the rectangular defects employed in the research and the actual corrosion conditions. However, within the existing spontaneous magnetic leakage theory models, the rectangular simplification is widely applied. This model simplifies the corrosion process of ferromagnetic materials as a loss in cross-sectional area. Several scholars have already verified the feasibility of this model in characterizing the corrosion state. (Literature [28] titled “Research on the Method of Predicting Corrosion Width of Cables Based on the Spontaneous Magnetic Flux Leakage.” Literature [33] titled “Quantitative Study on Corrosion of Steel Strands Based on Self-Magnetic Flux Leakage.”)

(2) As shown in Fig. 5, the electrochemical corrosion of the steel strand structure was adopted as the common method in the experimental research. Meanwhile, the localized corrosion mode could be regarded as the more severe than the naturally corroded defect, which is sensitive to the cable vibration state. The naturally corroded defect was composed of multiple irregular defects. So, the research findings in this study could provide valuable data reference for the complex stress states and defect forms in future study.

Point 3:

It is well-known from engineering practice that the region adjacent to the cable anchor is prone to severe corrosion. How effective is the proposed SMFL approach for this part? Are there any limitations or challenges that need to be addressed?

Response 3:

Thanks for your comment. Our responses are as follows:

Firstly, we do recognize the widespread issue of corrosion in the region adjacent to the anchorages of the cable. The spontaneous magnetic leakage detection method has proven to be effective in detecting corrosion near the anchorage area, but it also has certain limitations and challenges.

On one hand, the effectiveness of the spontaneous magnetic leakage detection method may vary depending on the specific cable structure and materials, necessitating further validation and optimization in practical applications. On the other hand, the anchorage cups in the anchorage area are also made of ferromagnetic materials, which can interfere with the leakage signals in the surrounding area during actual detection.

Given these limitations and challenges, we will continue to conduct research and experiments to continuously improve our spontaneous magnetic leakage detection method, aiming to enhance its applicability and reliability in corrosion detection in the vicinity of cable anchorages.

Point 4:

The work introduces a method for detecting corrosion defects using magnetic signals. However, the performance of this method may be affected by various environmental factors. Could the authors elaborate on what are the potential sources of interference or noise that can degrade the quality or reliability of the magnetic signals? How can these factors be mitigated or eliminated?

Response 4:

Thanks for your comment. Our responses are as follows:

In our method, there are indeed potential sources of interference or noise that could affect the quality or reliability of the magnetic signals, such as external magnetic field interference, electromagnetic noise, temperature changes, influences from metallic materials, effects of alternating loads, and so on.

To mitigate these interferences, we can employ shielding enclosures in the experimental environment to reduce the impact of external magnetic fields and electromagnetic noise. Temperature monitoring and control can be implemented to alleviate the influence of temperature fluctuations on the performance of magnetic sensors. Simulating the effects that cables may experience under alternating loads (such as vibration effects considered in this study) can help reduce the impact of these effects on magnetic signal acquisition.

It is worth noting that the leakage detection method may still encounter complex environmental conditions and interferences in practical applications. Therefore, we will continue to conduct experiments and validations to optimize the method and improve its reliability and effectiveness under different environmental conditions.

Reviewer 2 Report

1) The authors are advised to corroborate the detection of degree of corrosion as determined from SMFL curves with conventional detection methods. This will offer calibration to the method presented in the manuscript.

2) How sensitive the method would be to probe stress corrosion induced crack initiation or propagation?

3) What kind of magnetic sensor was used? Hall sensor or something different? The sensitivity of the sensor is required to be mentioned. Also please compare how the resolution of the sensor is placed against peak height.

4) Some basic material properties of the steel wire used should be indicated in the manuscript such as permeability, remnance, magnetic saturation, coercivity etc.

Author Response

Responses to Reviewer #2 Comments

Dear Reviewer:

Firstly, we greatly appreciate your comments and suggestions on our manuscript. Those comments are valuable and helpful for revising and improving our paper and the essential guiding significance to our studies. We have studied all the comments and suggestions carefully. Then we have made revisions that can meet your approval. The responses to your comments are presented as follows:

Point 1:

The authors are advised to corroborate the detection of degree of corrosion as determined from SMFL curves with conventional detection methods. This will offer calibration to the method presented in the manuscript.

Response 1:

Thanks for your comment. Our responses include two aspects:

(1) In this research’s experimental study, the actual corrosion ratio of the specimens plays a crucial role. The commonly used method to determine the actual corrosion ratio is through weighing. However, due to the long length and prior tensioning of the strand specimen, weighing the specimen itself was inconvenient. Therefore, in this experiment, the corrosion ratio of the steel strands was controlled using electrochemical corrosion and the empirical formula for the theoretical corrosion rate. The accuracy of this method has been validated in the literature [32] titled “Experimental Study on Corrosion of Unstressed Steel Strand based on Metal Magnetic Memory.” Therefore, the corrosion ratio results obtained through the application of Faraday’s law can be deemed reliable.

(2) Since the corrosion ratio of the steel cable calculated using Faraday’s law is considered reliable, the results in Section 4.3 of the manuscript, which assess the correlation between corrosion ratio detection and the SMFL curve employing an F-test, can also be regarded as reliable. The test results indicate a significant correlation between the characteristic index of the SMFL curve and the corrosion ratio. Consequently, for future studies, we will conduct additional experiments to optimize this index further, aiming to enhance its reliability in engineering practice.

Point 2:

How sensitive the method would be to probe stress corrosion induced crack initiation or propagation?

Response 2:

Thanks for your comment. Our responses are as follows:

The spontaneous leakage magnetic detection method has demonstrated a certain sensitivity in detecting crack initiation or propagation caused by stress corrosion. Previous studies have found that the spontaneous leakage magnetic detection method exhibits a certain sensitivity in detecting cracks induced by stress corrosion. This is because the magnetic field changes around the crack significantly affect the spontaneous leakage magnetic signals. By analyzing and comparing the magnetic signals under different conditions, we are able to detect crack initiation and propagation and understand the severity of the cracks.

However, its sensitivity can be influenced by various factors, including the size, morphology, and orientation of the cracks. Smaller and shallower cracks usually result in smaller changes in the magnetic signals, thus requiring sensors with higher sensitivity and resolution for effective detection. Additionally, factors such as material properties and testing conditions may also influence the detection results.

In the future, we will further validate the detection capability of the spontaneous leakage magnetic detection technology for cracks through experiments, optimize this method, and enhance its reliability and application value in engineering practice.

Point 3:

What kind of magnetic sensor was used? Hall sensor or something different? The sensitivity of the sensor is required to be mentioned. Also please compare how the resolution of the sensor is placed against peak height.

Response 3:

Thanks for your comment. Our responses are as follows:

The magnetic sensor used in this experiment is the Honeywell HMR2300 magnetoresistive sensor, with a measurement range of -2Gs~2Gs and a resolution of 67uGs. It collects 10~157 data points per second. This sensor can capture the magnetic field intensity components in three directions, namely Bx, By, and Bz.

Based on the experimental results, it was found that the maximum magnetic field intensity values obtained from the tested steel cables after corrosion did not exceed 1000mGs, within the range of the sensor’s measurement capability. The impact of vibrations can cause fluctuations in the magnetic signals near the peak values. However, these fluctuations fall within the resolution of the sensor. Therefore, this sensor can effectively observe the changes in the magnetic signals of the steel cables under vibration conditions. Additional information about the magnetic sensor is included in Section 3.3 (Lines 582-584) of the manuscript.

Point 4:

Some basic material properties of the steel wire used should be indicated in the manuscript such as permeability, remnance, magnetic saturation, coercivity etc.

Response 4:

Thanks for your comment. Our responses are as follows:

The experiment material was a smooth steel strand (1×7Φ5). The strand had a nominal diameter of 15.20 mm and a single length of 7 m. The chemical components and basic material properties of this steel strand specimens have been supplemented in Table 1 of Section 3.1 (Highlighted in yellow in the manuscript, lines 274~275). The manuscript also indicates that the relative permeability of steel strand μ=2000, and the saturation magnetization intensity Ms=1×106A/m (Lines 486-488).

Round 2

Reviewer 2 Report

The authors have implemented the suggested changes and responded appropriately to other questions. Therefore the manuscript can be accepted now.